# Brake Fluid Condition Monitoring by a Fiber Optic Sensor Using Silica Nanomaterials as Sensing Components

**DOI:** 10.3390/s24082524

**Published:** 2024-04-15

**Authors:** Mayza Ibrahim, Stanislav Petrík

**Affiliations:** Department of Advanced Materials, Institute for Nanomaterials, Advanced Technologies and Innovation, Technical University of Liberec, Studentská 1402/2, 46001 Liberec, Czech Republic; stanislav.petrik@tul.cz

**Keywords:** optical fiber sensor, electrospinning, silica nanofibers, silica aerogel, intensity modulated optical fiber sensor, refractive index of nanofibers, optical properties of nanofibers and aerogel, humidity, moisture sensors, air gap, sensitivity

## Abstract

In the automotive industry, there has been considerable focus on developing various sensors for engine oil monitoring. However, when it comes to monitoring the condition of brake fluid, which is crucial for ensuring safety, there has been a lack of a secure online method for this monitoring. This study addresses this gap by developing a hybrid silica nanofiber mat, or an aerogel integrated with an optical fiber sensor, to monitor brake fluid condition. The incorporation of silica nanofibers in this hybrid enhances the sensitivity of the optical fiber glass surface by at least 3.75 times. Furthermore, creating an air gap between the glass surface of the optical fiber and the nanofibers boosts sensitivity by at least 5 times, achieving a better correlation coefficient (R^2^ = 0.98). In the case of silica aerogel, the sensitivity is enhanced by 10 times, but this enhancement relies on the presence of the established air gap. The air gap was adjusted to range from 0.5 mm to 1 mm, without any significant change in the measurement within this range. The response time of the developed sensor is a minimum of 15 min. The sensing material is irreversible and has a diameter of 2.5 mm, making it easily replaceable. Overall, the sensor demonstrates strong repeatability, with approximately 90% consistency, and maintains uncertainty levels below 5% across specific ranges: from 3% to 6% for silica aerogel and from 5% to 6% for silica nanofibers in the presence of an air gap. These findings hold promise for integrating such an optical fiber sensor into a car’s electronic system, enabling the direct online monitoring of brake fluid quality. Additionally, the study elucidates the effect of water absorption on the refractive index of brake fluid, as well as on the silica nanomaterials.

## 1. Introduction

Brake fluid plays a crucial role in the functioning of the brake system and the overall safety of automobiles. The major brake fluids used today consist of the following constituents: ethylene glycol, polyglycols, silicone fluids, and isobutyl alcohol [1,2,3]. These constituents have the same hydrophilic characteristics due to their molecular O–H structure, which allows for dipole attraction of other hydroxy molecules through hydrogen bonding. Figure 1 illustrates how easily brake fluid can absorb water from its surroundings. Motor vehicle manufactures often recommended changing brake fluid annually or every two years. Some people commit to this recommendation, while others do not. The rate at which brake fluid absorbs moisture varies depending on factors such as the type of fluid, ambient humidity, the condition of the brake hoses and seals, and the vehicle’s mileage. Typically, brake fluid absorbs moisture at a rate of 1% or more per year of service. The presence of 2% water reduces the boiling point of DOT 4 (DOT stands for U.S. Department of Transportation) brake fluid by approximately 81 degrees Fahrenheit (45 °C) [4]. Brake fluid should be changed when it reaches its recommended life span, which is typically indicated when the water content reaches 3.5% [5].

Given that brake fluid is not isolated within a vacuum system in automobiles, it is susceptible to moisture infiltration. This moisture poses a significant threat to the effectiveness of brake fluid, as it accelerates corrosion within the brake system components. Additionally, moisture lowers the boiling point of the brake fluid, resulting in premature vaporization before it reaches its intended boiling point. Ultimately, this degradation can compromise the braking performance, potentially leading to reduced capacity or complete brake failure [6].

Currently, the evaluation of the quality of brake fluid is achieved through checking its boiling point. Currently, various methods are employed to estimate the boiling point of brake fluid, including the equilibrium reflux boiling point method, as well as electrical (specific conductivity, permittivity), and optical techniques (refractometry, spectrometry). The physical method for determining the equilibrium reflux boiling point is standardized and directly measures the brake fluid boiling point under controlled conditions. However, its practical application is limited due to its complexity and the requirement for a large sample size, potentially the entire volume of the brake fluid reservoir. Additionally, the sample heated during analysis cannot be reused due to chemical reactions occurring at high temperatures. 

On the other hand, electrical and optical methods offer non-destructive means of indirectly indicating the boiling point. The electric methods rely on measuring electrical conductivity and permittivity [4]. In a study by Barabas et al. [7], they compared conductivity and capacitance methods for estimating the boiling point of glycol-based brake fluids. They concluded that the capacitance method provides more accurate results, as the conductivity method is significantly influenced by water conductivity. Furthermore, Mogami [8] described a new method for measuring the boiling point of a small quantity of brake fluid using a thermocouple, with an accuracy of ±3 °C. These methods are considered to be physically time-consuming. Hence, an online method is essential to assess the quality of brake fluid. However, until now, a safer method for the online monitoring of brake fluid has not been available [9]. 

A few studies regarding online brake fluid monitoring have been reported [10,11]. Chuantong Wang et al. [10] proposed a method for online monitoring the condition of brake fluid. However, it relies on a capacitive sensing mechanism, which detects changes in permittivity due to water absorption. Nevertheless, these sensors primarily rely on electric methods, which pose safety concerns when dealing with a flammable substance. Optic methods offer a safer alternative; however, many optical sensors remain bulky, expensive, and complex. Consequently, there is ongoing research to incorporate optical fibers into sensing applications.

Fiber optic sensor technology has experienced rapid advancement over the past three decades, driven by innovations in the telecommunications, semiconductor, and electronics sectors. These advancements have led to significant reductions in the prices of optical components and have accelerated the development of optical fiber sensors [12,13]. Optical fiber sensors are capable of measuring a wide range of physical properties, including chemical changes, strain, electric and magnetic fields, pressure, temperature, displacement (position), radiation, flow, liquid level, vibrations, and light intensity. Optical fiber sensors offer numerous advantages over conventional electrical and electronic sensors. These include their small size and weight, enabling their use in otherwise inaccessible locations, as well as their capability for remote sensing. Additionally, optical fiber sensors exhibit resistance to radio frequency and electromagnetic interference. To further enhance their sensitivity, a layer of sensitive materials is applied to their surfaces.

Coating the surface of optical fiber sensors with a selective coating is a method used to enhance their sensitivity [14,15,16,17,18]. In the context of humidity and moisture sensors, the surface is coated with a traditional hygroscopic-sensitive material to improve water sensitivity [19]. The material’s refractive index changes in response to variations in humidity or moisture levels, thereby altering the power intensity of light within the optical fiber. Various methods exist for producing thin films, including vacuum evaporation [20], ion sputtering coating [21,22], sol–gel technology [23,24], layer-by-layer self-assembly (LBL) [25,26], supercritical drying [27,28,29], and the dip-coating method [30,31]. Among these techniques, electrospinning is noted for its simplicity and effectiveness in forming nanofiber layers, while the supercritical drying of gel is considered effective for producing highly porous structures. In this study, both silica nanofibers and aerogel were separately employed as coatings for the glass surface of the optical fiber.

Electrospinning demonstrates the remarkable capability of any material at the nanoscale, particularly in the realm of sensing materials. At the nanoscale, numerous features become easily accessible and even enhanced, such as the development of excellent mechanical properties, notably flexibility, high porosity, and a large surface area. These attributes increase the number of accessible sites for surface functionalization. From a sensor perspective, another crucial aspect of electrospinning is its ability to produce continuous nanofibers. This characteristic is vital, as sensors are typically integrated into specific measuring systems that involve analog-to-digital conversion. Therefore, a continuous and stable signal is necessary for accurate readings. In the context of humidity/moisture sensors, the nanofiber layer processed by electrospinning offers a significantly larger specific surface area compared to that of conventional coating films. This increased surface area enhances the sensor’s capability to absorb more water molecules, thereby improving its sensitivity to changes in humidity or moisture levels [16,32,33,34], while the exceptional properties of aerogels, including their remarkably high surface area, extensive porosity, and low density, make them highly attractive for use in sensing applications.

Inorganic materials, especially silica, are considered to be one of the most promising options for fabricating either nanofibers or aerogels due to its astonishing characteristics, such as low thermal conductivity and chemical inertness. Silica has been used in different applications such as electronics, medicine, cosmetics, food, filler applications, and consumer products [35,36,37]. Regarding the use of silica nanofibers in humidity sensing applications, Batool et al. [38] studied the effect of relative humidity RH on the dielectric response of SiO_2_ nanofibers. However, there is limited research on the influence of moisture on the refractive index of SiO_2_ nanofibers (NFs).

The initial precursor used to fabricate either silica nanofibers or aerogels typically involves the sol–gel procedure. There is a wide range of starting materials used to prepare the sol–gel precursors. Primarily, silicon alkoxides or water glass are commonly utilized. Alkoxides are popular precursors because they readily react with water. Although water glass is another cheaper alternative, the gels produced from this source are often very fragile and require purification before they can be utilized in aerogel production [39].

Synthesis of the silica aerogel can be categorized into three steps:Gel preparation: the silica gel is obtained from the sol–gel process.Aging the gel: the gel prepared in the first step is aged in its mother solution to strengthen the gel and minimize the shrinkage during the drying process.Drying the gel: during this step, the aerogel should be free of the pore solvent. To prevent the collapse of the gel structure, drying should be performed under supercritical conditions.

The drying process is critical and determines the structure and properties of the produced material. Supercritical drying produces an aerogel in which the pore liquid is removed above the critical temperature and critical pressure of the extraction fluid. However, drying at ambient pressure produces xerogel film. Freeze drying produces cryogel, as illustrated in schematic Figure 2.

Silica aerogels have attracted more attention due to their extraordinary properties, which provide them with a wide range of potential applications [40,41,42,43,44]. These properties can be summarized as their high specific area (500–1200 m^2^/g), high porosity (80–99.8%), low density (~0.003 g/cm^3^), low thermal conductivity value (0.005 W/mK), ultra-low dielectric constant (k = 1–2), and low index of refraction (~1.05). The proposed applications of aerogels are many and varied, including their use in thermal insulation, acoustic barriers, and supercapacitors. Other applications of silica aerogels include their use in shock wave studies at high pressure [45,46,47,48,49], inertial confinement fusion (ICF) [50,51,52], and radio luminescent devices [53,54,55]. Additional applications remain unrealized, as yet [38].

In this work, silica in the form of nanofibers or aerogel was tested to check its validity in terms of changing its optical characteristics due to absorbing water molecules at a very low percentage, ranging from 0 to 6% weight percent (wt%) in brake fluid. This study also examines the impact of water absorption on the refractive index of SiO_2_ nanofiber mat and silica aerogel, while employing optical fiber as the transduction mechanism. The novelty of this research lies in employing silica in the form of nanofiber mat and aerogel as a water absorbing materials. The absorption of water molecules alters the optical properties, which can be detected by the optical fiber sensor. Another novelty of this study is the introduction of an air gap between the glass surface of optical fiber and the sensing material, which significantly enhances the sensitivity. The developed sensor can be easily integrated into a car’s electronic system to detect the moisture level (water content) in the brake fluid. Furthermore, the prepared sensor can be industrialized by incorporating the sensing material into a special adaptor which can be easily replaced by car driver. The study result is promising and paves the road for other researchers to explore this potential application of silica porous materials in humidity/moisture sensing applications.

## 2. Materials and Methods

### 2.1. Chemicals

The chemicals used include TEOS (tetraethyl orthosilicate, 98 wt.%, Sigma-Aldrich, Merck Life Science spol.s.r.o., Praha, Czech Republic), TMOS (tetramethyl orthosilicate, 99.7% Alfa Aesar, Thermo Fisher Erlenbachweg 2, Kandel, Germany), ethanol (99.9 wt.%, Penta CZ, Katovice, Czech Republic), methanol (99.8% VWR, CZ, Stříbrná Skalice, Czech Republic), distilled water, and hydrochloric acid (2 mol/L).

### 2.2. Characterisation Methods

To investigate the chemical bonds of the silica nanofibers, IR spectra were recorded on a Nicolet iZ10 (Thermo Scientific, Waltham, MA, USA) FTIR spectrometer with a DTGS detector using the ATR technique, with a spectral resolution of 4 cm^−1^ and a spectral range of 4000–400 cm^−1^.

The structural analysis of the silica nanofibers and the silica aerogel samples was performed using a UHR-FE-SEM Zeiss Ultra Plus 3 device, equipped with energy dispersive X-ray analysis (EDX) with SEM-LEO S1430 VP from M/S LEO Electron Microscopy Ltd, Cambridge, UK and UHR FE-SEM Carl Zeiss ULTRA Plus, Carl Zeiss Meditec AG, Jena, Germany, to investigate the chemical composition of the silica nanofibers. The samples were fixed on aluminium stubs using double-sided carbon tape; the samples were not modified or metal-coated before the analysis. Gentle beam conditions, including a low accelerating voltage of 1 kV and low probe currents around 3 pA, were used to obtain images because of the beam sensitivity and low average atomic number of the observed materials. EDX analysis was performed at 10 kV accelerating voltage. An Oxford X/Max20 (SW AZtec 3.3) EDX detector was used to collect the spectrum. 

The specific surface area of the aerogel sample was evaluated through adsorption/desorption isotherms, measured with nitrogen at −195 °C (N_2_, Linde, 99.999% purity) using ASiQWin software (version: 5.21) from (Quantachrome Instruments, AutoSorb iQ). Prior to the measurement, the sample underwent degassing under vacuum at 300 °C for several hours. The specific surface area was calculated using the Brunauer–Emmett–Teller (BET) method. Additionally, the volume and pore size distribution were determined based on the desorption curve utilizing the density functional theory (DFT) method (cylinder. pore, NLDFT equilibrium model).

An optical power-meter, model XL-FMS130KU, was used as a laser source and a photodetector. The resolution of the laser is 0.05 dB, while for the uncertainty of the detector is 5%. A Kruss DR201-95 refractometer was utilized to measure the refractive index of the brake fluid.

### 2.3. Nanofibers Samples Preparation

The spinning solution for the SiO_2_ nanofibers was prepared as follows:

TEOS (3 mL) and ethanol (3.3 mL) were mixed and stirred for 15 min in a baker at ambient temperature. A solution of ethanol (2 mL), 2 M HCL (80 µL), and distilled water (600 µL) was added to the mixture under continuous stirring for 30 min. The solution was heated up to 200 °C for 30 min to distill approximately 4 mL of ethanol, or until the total volume became 5 mL.

TEOS has the chemical formula Si(OC_2_H_5_)_4_ or Si(OR)_4_, where R represented the ethyl group (C_2_H_5_). In the preparation of the silica solution for electrospinning, the sol–gel technique was utilized. A sol, by definition, is a suspension of nanoscale solids ranging from 1 nm to 1 µm in a liquid phase [56]. This suspension is achieved through the polymerization reaction of a metal source to colloidal metal oxides using an acid and/or base catalyst in a wet medium. The overall reaction involves the reaction of one molecule of TEOS with two equivalents of water, leading to the formation of silica in its fully condensed form, along with four equivalents of ethanol, as shown in Equation (1) [57]. The polymerization frequently involves the hydrolysis of alkoxy groups in TEOS, a process that can be occur under either an acidic or basic condition.
(1)$$\mathrm{Si}\;{\left(\mathrm{OR}\right)}_{4}+2{\;H}_{2}O\to {\mathrm{Si}\;O}_{2}+4\;R-\mathrm{OH}$$

Intermediate species such as [(OR)_2_–Si–(OH)_2_] or [(OR)_3_–Si–(OH)] may result as products of partial hydrolysis reactions. Polymerization is associated with the formation of a one-, two-, or three-dimensionl network of a siloxane (Si–O–Si) bond, water, and ethanol.

For the preparation of nanofibers mats, the electrospinning conditions were adjusted as follows: the electrode distance is 140 mm, the substrate speed is 0.2 mm/s, the head speed is 350 mm/s in 500 mm, the voltage is 57 KV, the process air flow is 80–120 m^3^/h, the air humidity is 33%, and the laboratory temperature is 23 °C. A baking paper, without silicone, was used as a substrate for all the produced nanofibers. Nanofibers mats were then heat treated at 120 °C for 30 min.

### 2.4. Silica Aerogel Preparation

For the preparation of the sol–gel solution, TMOS (16.56 g) was mixed with methanol in one container, and the water was mixed with 2 M NH_4_OH in a second container. The chemicals were mixed together in the proper molar ratio of TMOS:MeOH:H_2_O:NH_4_OH (1:4:4.5:0.017) and transferred to the smaller glass tubes. After approximately 10 min, the samples became rigid alcogels. To each alcogel sample placed in the glass bottle, 15 mL of methanol was added to prevent shrinkage under ambient drying, and the solvent covered the samples. Over a month, the gels, contained within sealed glass bottles, underwent complete immersion in various solvent baths (methanol and acetone). Throughout this process, any remaining unreacted water diffused out of the gel, allowing the network to strengthen [58,59].

For the preparation of the silica aerogel, the following conditions were adjusted: the samples were placed in a tubular pressure vessel and covered with the remaining solvent. Subsequently, the vessel was filled with liquid CO_2_ at laboratory temperature and with the corresponding equilibrium pressure for CO_2_ (approx. 70 bar). The samples were left overnight (16 h). The pressure was increased to 100 bar (to ensure a 100% liquid phase), and the excess liquid solvent was slowly drained until only a small amount remains, which takes approximately 30 min. After that, the flow was stopped, and the samples were held for 2 h for stabilization and exchange of the solvent with CO_2_. A flow rate of 1 L/min. (gas phase) was then set for 30 min. Cycles of 2 h standing/30 min rinsing were repeated a total of three times, and after the last standing, the temperature was increased to 45 °C, and the pressure was kept at 100 bar (to ensure sc-CO_2_). After the temperature had stabilized, a gradual depressurization of the reactor was set at 0.5 bar/min, with a constant compliance temperature at the critical point (45 °C). After several hours of depressurization, the heat was turned off, and the samples were removed from the reactor (usually the next day).

### 2.5. Sensor Fabrication

The developed sensor is based on the intensity-modulated optical fiber sensor principle (IM-OFS) in reflection mode. The essential building blocks of the fabricated sensor are depicted in Figure 3a.

A multi-mode optical coupler (Y coupler) with core and cladding diameters of 62.5 and 125 µm was used with a fiber-optic connector as a physical contact connector (FC/PC). An XL-FMS130KU optical power meter was used as a laser source, as well as a photodetector. The laser with a wavelength of 1310 nm was coupled into one branch of the optical coupler for transmission to the FC/PC. A special adaptor (Figure 4b) is attached to the connector, where porous material is attached. This adaptor is then immersed into the measurands. Different proportions of light will be reflected and coupled back with the same optical fiber, due to changing of the measurands. These signals are then transmitted to a photodetector. Both the laser and the measurement of the reflected power were set at the same wavelength of 1310 nm. Throughout the experiments, the optical fiber remained at a right angle by fixing it on a 3D-printed frame, as shown in Figure 3b. In our case, the measurand is the quality of the grade DOT 4 brake fluid. The quality of DOT 4 was represented by its weight percent of water. The relative reflected power was recorded as a function of the water concentration in DOT 4. The reference point for reflected power was set to the value obtained from pure DOT 4 with 0% water content in all measurements.

A round cutter with a diameter of 2.5 mm was used to cut the nanofibers, while for the silica aerogel, the sample was manually adjusted to fit onto the adaptor. Either silica nanofibers or aerogel was inserted into the adaptor to rest on surface (B), as shown in Figure 4a. This adaptor served to securely hold either the silica nanofibers or the aerogel, allowing for direct contact or the creation of an air gap between the FC/PC connector (surface A in Figure 4a). This gap was established by withholding the key plug from its sheath, as depicted in Figure 4b, resulting in an air gap with a length of approximately 0.5 to 1 mm. A cross section of both the adaptor and the connector is shown in Figure 4a. Several adaptors were procured. A 2.5 mm piece of typical sensing materials was previously cut and inserted into these adaptors for easy exchange with the wetted adaptor between measuring cycles. The tip of the sensor was routinely cleaned with water and thoroughly dried after each cycle to ensure readiness for subsequent cycles. The sensor was calibrated by providing predetermined samples with a known water percent. The reflected power was recorded for each sample. The data was plotted on a calibration curve. The sensitivity is determined by the slope of the calibration curve. This cycle was repeated two days a week for two months. Uncertainty in the percent was expressed as the coefficient of variation (CV).

## 3. Results

### 3.1. Characterization of Electrospun Nanofibers Mat

A silica nanofibers mat was produced, with a thickness of 129 ± 28 µm. The morphology of the SiO_2_ nanofibers mat is characterized in images prepared by scanning electron microscopy. Figure 5 shows the scanning electron microscope (SEM) image of the pure SiO_2_ nanofibers associated with its fiber diameter distributions.

The average diameter of SiO_2_ fibers is 143 ± 42 nm. The compositional analysis confirmed their fabrication to be nearly stoichiometric, with silicon (Si) and oxygen (O) constituting 51.8% and 48.2%, respectively. FTIR confirms the creation of SiO_2_ through the formation of the siloxane Si–O–Si bonds at 1067 and 797 cm^−1^, which become more intense after heat treatment, as shown in Figure 6. The disappearance of the hydroxyl group at 3315, 1633, and 945 cm^−1^ could be attributed to removing the traces of moisture contained in the sample.

### 3.2. Aerogel Characterisation

Figure 7a depicts the SEM morphology of the silica aerogel, while Figure 7b presents the N2 adsorption/desorption isotherm and the pore structure analysis conducted using BET and DFT methods for the tested silica aerogel samples.

The pores have a broad size distribution, with an average diameter of 22 nm (shown in the inset of the adsorption/desorption curve). The tested sample is defined by a classic final saturation plateau and exhibits a type-IV isotherm with an H3 hysteresis loop corresponding to slit-shaped pores, according to IUPAC classification [60], that are the result of mesoporous material. The BET surface area and the pore volume of the silica aerogel are 810 m^2^g^−1^ and 3.54 cm^3^g^−1^, respectively.

### 3.3. Characterization of the Developed Sensor

In the first step, the developed sensor was tested without any functional material. The sensing element was the glass tip at the distal end of the optical fiber. The data in Figure 8 show the relationship between the percentage of water content (from 0 to 6 weight %) in DOT 4 and the reflected power in dB.

The sensitivity of the developed sensor when using the glass tip as a sensing media is 0.06 dB/% water content, as shown in Figure 8. This sensitivity is increased to −0.2 dB/% water content after applying SiO_2_ NFs as a sensing media, as shown in Figure 9a.

The sensitivity increases fivefold to equal a −0.3 dB/% water content when an air gap is introduced between the glass tip of the optical fiber and the SiO_2_ NFs, with a higher R square of 0.98 (Figure 9b). Experimental results have shown that the sensitivity of the glass surface is enhanced by 3 and 5 times when nanofibers are applied, without an air gap and with an air gap between the glass surface of the optical fiber and the nanofibers, respectively.

The sensitivity of the glass is enhanced by 10 times by applying silica aerogel with an air gap, as shown in Figure 10, while the sensitivity does not significantly change when there is no air gap.

## 4. Repeatability of the Sensor

Uncertainty in regards to percent for each water increment is displayed in the Table 1 for both the silica nanofibers and the silica aerogel with an air gap.

The sensor exhibits up to 5% uncertainty in detecting water content exceeding 5% in brake fluid when coated with silica nanofibers. For silica aerogel coating, the uncertainty extends to the range of water content in brake fluid exceeding 3%, also with up to 5% uncertainty. These measurements are taken in the presence of an air gap. Despite using a bundle of specially manufactured optical fiber couplers with a diameter of 2.5 mm to increase the interaction volume of the silica nanomaterials with light, achieving repeatability without the air gap posed challenges. The experiment resulted in approximately 40% repeatability, indicating that out of 100 repeated trials, only around 40 produced similar results. Furthermore, within these 40 trials, the variation remained high, contributing to significant uncertainty. Additionally, the use of a powerful optical light source was necessary to support signal transmission into the fiber and to withstand the increased power loss during the submersion of the optical fiber into the brake fluid. Therefore, this bundle was abandoned after testing.

## 5. Discussion

The proposed sensor operates on the principle of modulating the intensity of the reflected light. When light strikes the interface between two distinct media, it can undergo both reflection and refraction. The portion of incident power reflected from the interface is known as reflectance or reflectivity, which is quantified by the power reflection coefficient R. Conversely, the portion of light that refracts into the second medium is termed transmittance or transmittivity, represented by the power transmission coefficient. Upon encountering the surface of the porous structure, whether nanofibers or aerogel, the incident waves are divided into three segments: reflected, absorbed, and transmitted waves [61,62], as illustrated in Figure 11.

The Fresnel equations describe the reflection and transmission of light when light encounters an interface between different optical media, and they do not account for the attenuation of a wave in an absorbing media. The sensing mechanism of the proposed sensor relies on the detection of the variations in reflective power due to water absorption. The absorption of water significantly alters the refractive index of both brake fluid and the silica nanomaterials, consequently influencing the reflected power at the interface. Since the reference point was set to the reflected power from a pure DOT 4, it is expected that the coated material will exhibit varying degrees of water absorption Consequently, variations in reflected power may be detected at the interface. Based on Fresnel equation, the reflectance will vary according to the refractive indices of both media. In the case of normal incident, as in our case, the reflectance simplifies to Equation (2)
(2)$$R={\left|\frac{n_{\mathrm{core}}-n_{s}}{n_{\mathrm{core}}+n_{s}}\right|}^{2}$$
where n_core_ and n_s_ are the refractive indices of the core of the optical fiber (OF) and the surrounding media, respectively. The intensity of the reflected power depends on the optical properties of the core of the OF and the surrounding media. In the case of a plain core OF, without nanofiber incorporation, the variation in reflected power is influenced by the refractive indices of the DOT 4 samples. This relationship is governed by the Lorentz–Lorenz formula [63], which correlates a material’s refractive index with its density and polarizability, as described in Equation (3)
(3)$$\frac{{n_{m}}^{2}-1}{{n_{m}}^{2}+2}=\frac{4{\pi N}_{A}\gamma \rho }{3M}$$
where n_m_ is the medium refractive index, M is the medium molecular weight, ρ is the density of the medium, N_A_ is the Avogadro constant, and γ is the medium polarization index. As the water content increases in the brake fluid, the brake fluid becomes less dense (with lower refractive index). As a result, the reflected power will gradually increase as indicated from the result of using only the glass, as shown in Figure 8. This explains the positive trendline in the case of the glass sample.

The explanation regarding the modification of the glass surface of OF with a porous structure, whether it be nanofibers or aerogel, is somewhat complex due to the heterogeneous nature of the porous structure (including factors such as fiber diameters, distribution, porosity, etc.), and the interaction of the incident light waves, with a wavelength comparable to the diameter of nanofibers. Silica exhibits hydrophilic properties due to the presence of silanol (Si–OH) on the surface of the nanofibers or aerogel. Generally, these hydroxyl groups enhance their ability to absorb water through hydrogen bonding, as illustrated in Figure 11. In the case of coating the core of the glass fiber with silica aerogel, the refractive index of the silica aerogel is primarily dependent on its bulk density, as indicated by Equation (4)
(4)n=1+kρ
where k represents a coefficient depending on the light wavelength [64]. The values of k at different wavelengths can be found in Ref. [65]. This correlation dated back to the work of Poelz and Riethumuller [66]. Several researchers have validated this theory [67,68,69]. P. wang et al. [69] studied the effect of the absorbed water on the refractive index of aerogel. Their study theoretically and experimentally confirms that the absorbed water causes an increase in the refractive index of aerogel. As the refractive index increases, the amount of reflected power is expected to decrease, according to the Fresnel equation. This explains the negative trendline observed in the sensor output of the sample using silica aerogel.

When modifying the glass surface of OF with nanofibers, the refractive index of the nanofibers is influenced not only by the material of the electrospun fiber, but also by the presence of absorbed water and air within its pores. As water is absorbed, the pores become filled, hence suppressing the air gaps, which have a lower refractive index than do water molecules. Therefore, this could contribute to an increase in the refractive index of the nanofibers, allowing more leaky modes to escape from the optical fiber. This explains the decrease in reflected power as water content increases in DOT 4, as shown in Figure 9a,b.

The decrease in reflected power in the samples employing both silica aerogel and nanofibers can also be explained by polarization loss caused by water absorption. Water molecules are the result of the higher negativity of the oxygen ions compared to the hydrogen ions. As a result, when water is absorbed by nanofibers or aerogel, it increases polarization loss. This means that the nanoporous structure absorbs more electromagnetic energy. Consequently, the reflected power decreases.

The sensitivity of the glass surface of the optical fiber is observed to increase with the presence of an air gap between the glass and the porous structure. This enhanced sensitivity of the developed sensor may be related to the numerical aperture (NA) and the acceptance angle of the optical fiber (θ_a_). NA is a property of the optical fiber and is independent of the refractive index of the surrounding media around the fiber; NA= $\sqrt{n_{core}^{2}-n_{cladding}^{2}}$; the NA of the utilized optical fiber is 0.275 [70]. NA is the sine of the half of the acceptance angle in a vacuum. The half of the acceptance angle (θ_a_) = ${sin}^{-1}\frac{1}{n_{s}}\;NA$; where n_s_ is the refractive index of the surrounding media. For air and pure brake fluid, the acceptance angles were calculated to be 32° and 22°, respectively, with the refractive index of pure brake fluid measured as 1.4429 using a Kruss DR201-95 refractometer. Applying trigonometry, the diameter of the cone (D) can be calculated: D = 2 × height of the air gap × tan (θ_a_/2). Thus, D ≈ ranges from 0.3 mm to 0.57 mm, as illustrated in Figure 12. Consequently, the enhanced sensitivity may be attributed to the fact that acceptance angle is significantly larger in air compared to that in a medium with a refractive index exceeding 1, thereby increasing the volume of the porous material interacting with the light exiting the optical fiber end. Additionally, because of the significant difference in the refractive index between the fiber glass core and the air, multiple reflections can occur in the air gap, resulting in the reflection of some optical power back into the interaction with the porous material/liquid media. Figure 12 illustrates the interaction volume for both cases.

This study reveals that the sensitivity of aerogel exceeds that of nanofibers. This difference may be attributed to the higher porosity of silica aerogel, which can range from 80% to 99.8% [4,5,6,7,8], compared to that of silica nanofibers, which typically exceed 80% porosity [71,72,73]. To the best knowledge of the authors, there is no study which uses the same sensing mechanism to detect the water content in brake fluid. Furthermore, there is a notable gap in research regarding the comparison of sensitivity between silica nanofibers and silica aerogel, whether in utilizing this specific sensing mechanism or in other humidity sensing mechanisms.

The response time of the developed sensor is at least 15 min. It is better to record the measurement after 1 h. This delay could be attributed to the time required for water molecules to diffuse into the porous structure, potentially involving either the physical or chemical absorption of water molecules.

The sensor demonstrates a satisfactory repeatability of 90%, and the remaining 10% could be attributed to inherent microcracks in the sensor material that may develop due to excessive force applied during installation. The sensors were developed, modified, and tested over two years, with different batches of brake fluid from the same brand type and with different optical fiber couplers as well. The sensor exhibits up to 5% uncertainty in detecting water content exceeding 5% in brake fluid when coated with silica nanofibers. For silica aerogel coating, the uncertainty extends to the range of water content in brake fluid exceeding 3%, also with up to 5% uncertainty. These measurements are taken in the presence of an air gap.

The durability of the sensor relies mainly on the durability of the optical fiber cables used. For practical use, the cables should be capable of enduring the rigorous routine changing of the adaptor that may arise. However, during the period of measurement, the durability of the optical cable was not a problem at all. The exchange of the optical fiber was due to scratches which may develop on its tip due to improper handling.

The limitation of adapting such a method in individual vehicles is the effect of vibration on the intensity of the light. However, manufactures of optical fiber cables must develop more durable optical cables, capable of enduring rigours daily use, while maintaining acceptable performance tolerance. Some manufacturers have already introduced optical fibers engineered for enhanced durability, specifically designed to withstand environmental factors such as vibration, mechanical stress, and other challenges inherent in communication networks and automotive sensing applications [74].

The effect of the cross sensitivity of the proposed sensor to temperature has been investigated. The range of the refractive index n in brake fluid span from about 1.4447 at 0% water to about 1.4394 at 7% water. The average value of the temperature coefficient of brake fluid is −0.000388416/°C, while the absolute value of the refractive indices at 20 °C lie within the range between about 1.44721 for DOT 4 brake fluid and about 1.44344 for DOT 3 [75]. The refractive indices of DOT 4 were measured for different concentrations of water. The refractive index is decreased by 0.0006 for each 1% increase in water content. Suppose the maximum temperature of the fluid is 90 °C on a hot summer day; that means that the refractive index will experience a total increase of 0.027. This value is very high when compared to the sensitivity of the glass surface, which is 0.06 for each 1% increase in water content. Due to the very low sensitivity of the glass, it is necessary to provide a temperature compensation. However, the increased sensitivity of the sensing material compensates for the temperature increase, especially in regards to the silica aerogel.

While only one brand of brake fluid has been tested in this research, it is believed that the proposed sensor can be calibrated for use with all different brands of brake fluids. This belief is based on the sensor’s higher sensitivity compared to that of glass. Different brands of brake fluid vary in their refractive index values at a level of the third decimal place [75]. This slight variation impacts the sensitivity of glass, which operates at the second decimal place with respect to a 1% increase in water content. However, the aerogel’s higher sensitivity, which is ten times that of glass, can compensate for the small variations in refractive index values among different brands of brake fluids, allowing the sensor to be effectively calibrated for use with any brand.

The proposed sensor offers environmental benefits by reducing the unnecessary waste of hazardous brake fluid that occurs when brake fluid is changed prematurely during routine inspections conducted every year or every two years. Brake fluid is inherently toxic and corrosive, making it a significant threat to both human health and the environment. With most of the brake fluid containing a glycol ether base, they can irritate skin upon contact and cause severe harm if digested. Furthermore, brake fluid is classified as a hazardous waste not only for humans, but also for soil, waterways, and plants. Proper disposal is not only recommended, but essential for safety and environmental protection. On the other hand, there are still some people who neglect to change their brake fluid regularly until it reaches critical levels. The hygroscopic nature of brake fluid allows water to accumulate inside, leading to rust in the brake lines and callipers, potentially damaging their performance. Therefore, implementing online monitoring of brake fluid could address these issues by reducing the unnecessary hazardous waste from premature replacements and mitigating the negative consequences of delayed replacement. Moreover, the sensor enables the real time, 24-h monitoring of brake fluid, allowing drivers to promptly schedule fluid changes when necessary.

This study provides a proof of concept and explores the potential integration of such a method into the monitoring system of vehicles, especially considering that after years of development, fiber optic networks are finally appearing in luxury automobiles. Automotive engineers first began seriously considering the use of fiber optics over two decades ago, initially aiming to employ them to prevent electromagnetic interference from affecting the operation of early electronic systems, like antilock brakes. Designing an electronic circuit to support both the light source and the photodetector could be a starting point for future research. This circuit could connect to the car’s electronic system to alert the driver to change the brake fluid, as the sensor design consists of three components—multimode fiber cables, an adaptor holding the sensing material, and the sensing material. The size of the fiber cable is just 2.5 mm, hence maintaining a compact size for use with a normal cable in cars. Consequently, the entire design can resemble normal cables used in cars. The authors plan to enhance the detection system in several ways, i.e., introducing an alarm triggered at a specific water content of the brake fluid, exploring the influence of porous material thickness and consistency on sensor performance, incorporating hydrophilic dopants like alkali metals and inorganic salts into silica aerogel, and investigating the reproducibility of the proposed sensor.

## 6. Conclusions

An intensity-reflected hybrid optical fiber sensor, with SiO_2_ nanofibers and silica aerogel, was fabricated and tested, resulting in the sensitivity of the glass being enhanced by 3, 5, and 10 times in the case of silica nanofibers, without and with an air gap, and silica aerogel with an air gap, respectively, to monitor the condition of the brake fluid. The air gap was adjusted to range from 0.5 mm to 1 mm. The measurement was held within this range, without any significant difference noted within this range. Although the sensing material is non-reversible, its small dimension of 2.5 mm diameter allows for easy replacement. Overall, the sensor demonstrates strong repeatability, with approximately 90% consistency, and maintains uncertainty levels below 5% across specific ranges: from 3% to 6% for silica aerogel and from 5% to 6% for silica nanofibers. While the reproducibility data is currently unavailable, it will be a focus of investigation in future research to ensure the reliability and validity of the findings. The sensor offers environmental benefits by enabling the real-time, 24-hour monitoring of brake fluid, facilitating prompt fluid changes, and reducing the production of hazardous waste from premature replacements. Additionally, it such monitoring helps to mitigate the negative consequences of delayed brake fluid replacement. Therefore, the developed sensor can be easily scaled for industrial use by integrating the sensing material into a specialized adapter that can be readily replaced by the user.

## Figures and Tables

**Figure 1 sensors-24-02524-f001:**
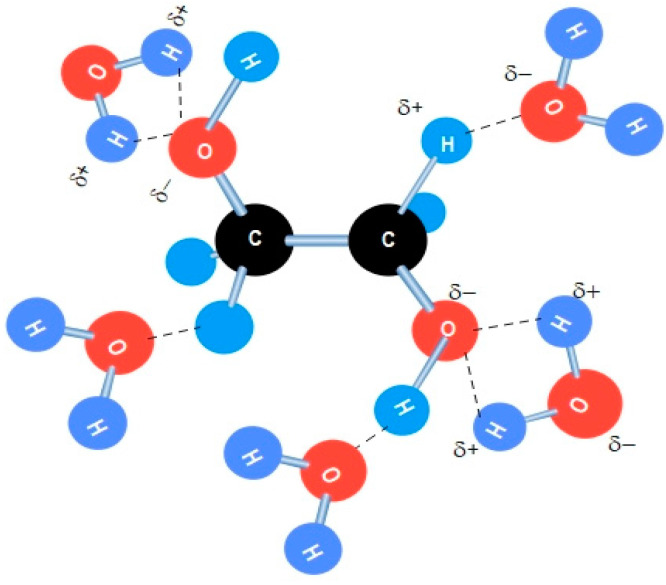
Illustration of hydrogen bond formed between ethylene glycol and water.

**Figure 2 sensors-24-02524-f002:**
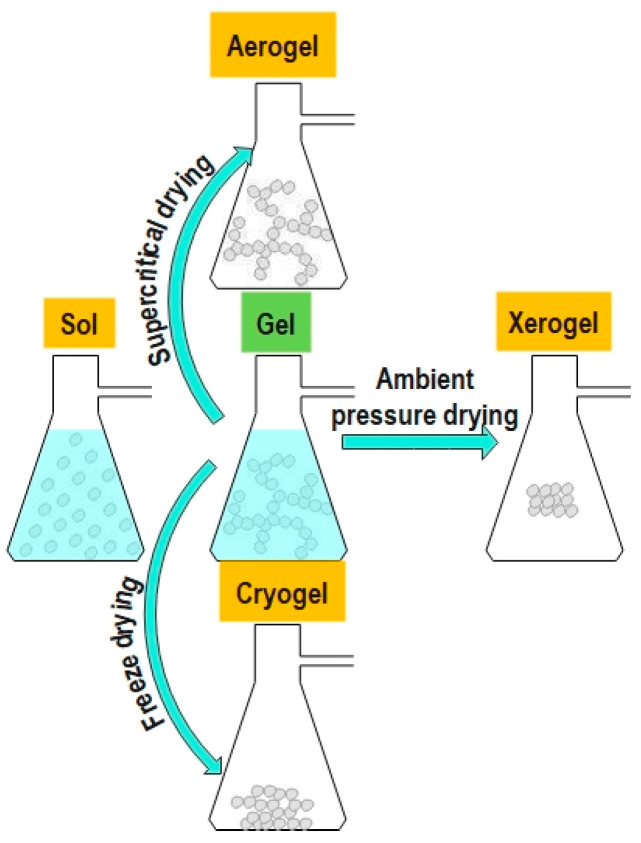
Schematic representation of the production of three different structures (aerogel, xerogel, and cryogel) using different drying techniques.

**Figure 3 sensors-24-02524-f003:**
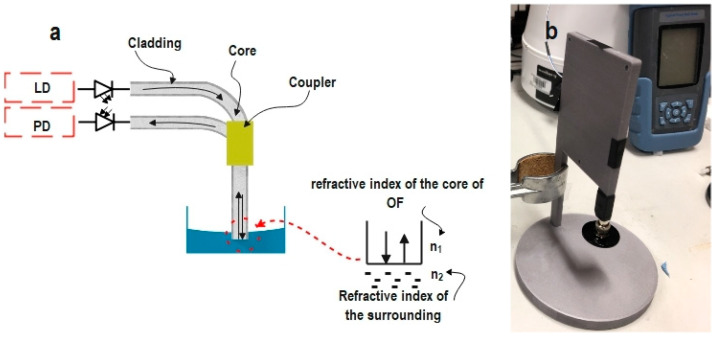
(**a**) Schematic representation of the operating principle of the sensor; (**b**) a photo of the developed sensor (PD—photodiode, LD—laser diode).

**Figure 4 sensors-24-02524-f004:**
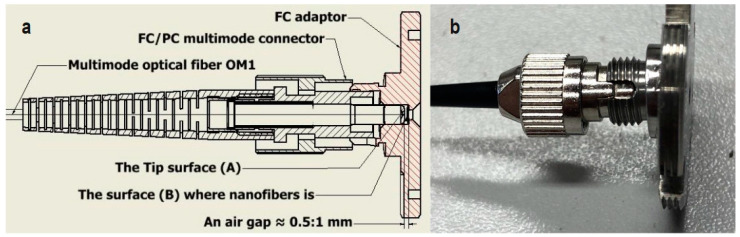
The adaptor used, with the FC/PC connector featuring the air gap, as illustrated in a half cross-section (**a**), while (**b**) is an image of this combination.

**Figure 5 sensors-24-02524-f005:**
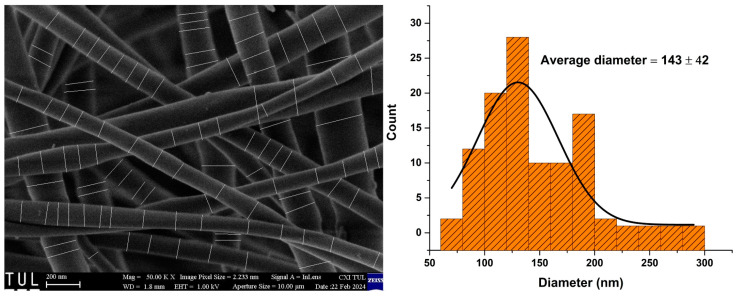
Scanning electron microscopy (SEM) images of SiO_2_ nanofibers and their fiber diameter distributions.

**Figure 6 sensors-24-02524-f006:**
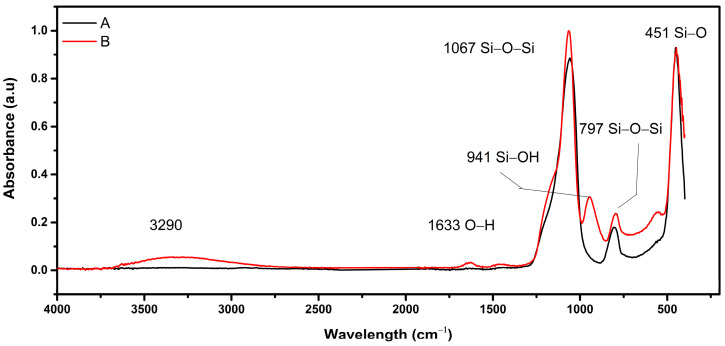
Normalized FTIR spectrum of SiO_2_ electrospun nanofibers at a peak of 451 cm^−1^, before (**B**) and after (**A**) heat treatment.

**Figure 7 sensors-24-02524-f007:**
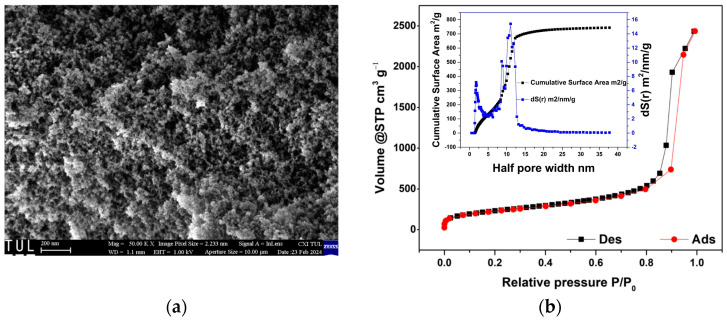
(**a**) SEM image of the silica aerogel cross section, and (**b**) N2 adsorption/desorption silica aerogel isotherms.

**Figure 8 sensors-24-02524-f008:**
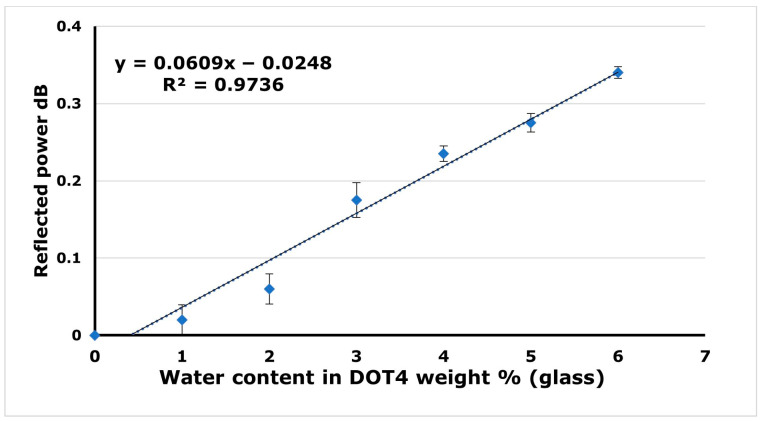
Sensor response in the case where the sensing media is glass only.

**Figure 9 sensors-24-02524-f009:**
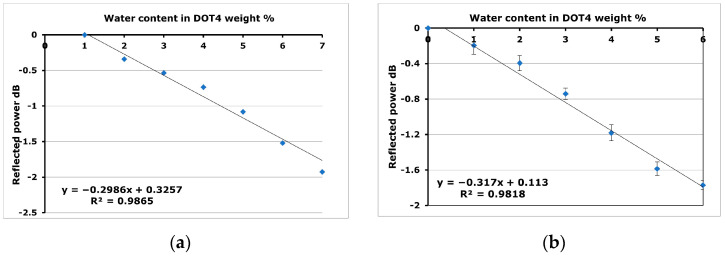
The response of the sensor when the sensing media is silica nanofibers without a gap (**a**) and with a gap (**b**) between the nanofibers and the glass, respectively.

**Figure 10 sensors-24-02524-f010:**
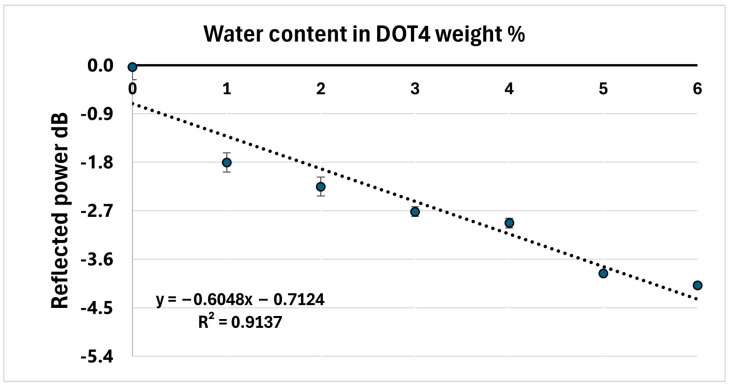
The response when the sensing media is silica aerogel.

**Figure 11 sensors-24-02524-f011:**
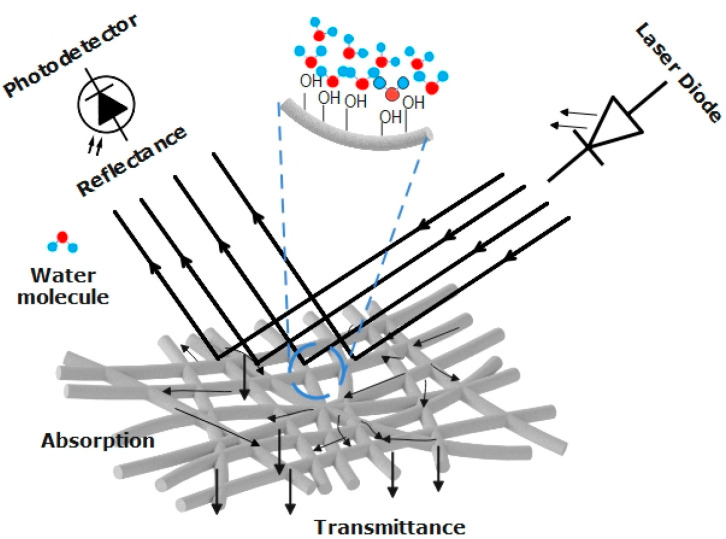
Schematic representation of the incident waves interacting with the porous structure.

**Figure 12 sensors-24-02524-f012:**
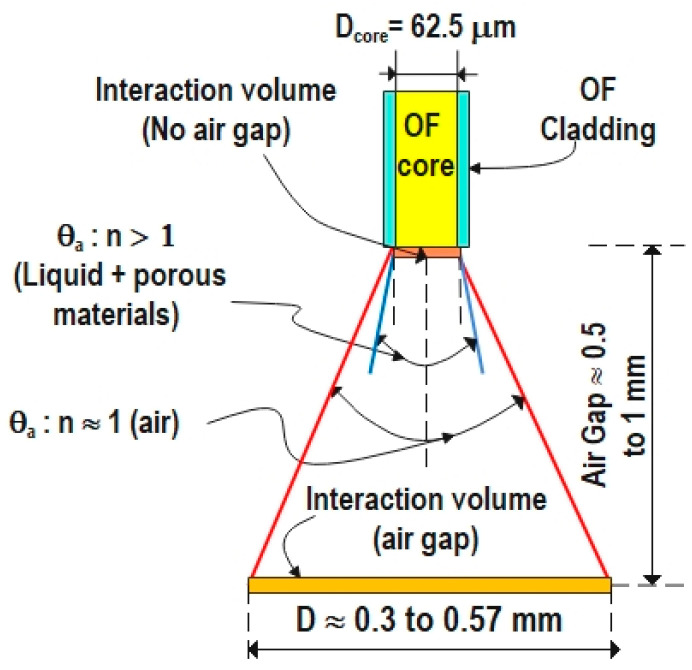
Acceptance angle of an optical fiber in air and a media with a refractive index higher than 1, respectively, and the interaction volume in the case of direct contact or with an air gap between the optical fiber and nanomaterial/liquid media.

**Table 1 sensors-24-02524-t001:** Uncertainty of the proposed sensor in terms of the coefficient of variation for silica nanomaterials at different water content levels in brake fluid, from 1% up to 6% weight percent.

Water content wt%	1%	2%	3%	4%	5%	6%
Sensor uncertainty (silica nanofibers)	51%	21%	8.7%	7.6%	4.7%	2.8%
Sensor uncertainty (silica aerogel)	9.8%	7.8%	3%	3%	0.9%	1.3%

## Data Availability

Data are contained within the article.

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
