# Peer review of "Brake Fluid Condition Monitoring by a Fiber Optic Sensor Using Silica Nanomaterials as Sensing Components"

_sensors, 2024, doi:10.3390/s24082524_

Round 1

Reviewer 1 Report

Comments and Suggestions for Authors

Comments on the Quality of English Language

Author Response

Dear reviewer, 

Thank you for your valuable questions, which have enabled me to further investigate and extract insights from my results, consequently improving the quality of the manuscript. Your input is greatly appreciated.

Author

Reviewer 2 Report

Comments and Suggestions for Authors

1. The title of this work "Brake fluid condition monitoring by fiber optic sensor with porous silica nanomaterials" is rather confusing. I thought it was a fiber made of porous silica. Moreover, microfibers are not a porous nanomaterial. Only the aerogels could fit within this concept

2. In addition, the authors only compared only against themselves. I mean, they just compared their system in three conditions: glass, microfibers and silica aerogel. Without a connection with the state of the art it is not possible to say if it is a good sensor

3. What about repeatibility and reproducibility of their sensors? How the authors tested the sensors several times in diffierent days? Can they produce a sensor that is similar to the previous one? Durability?

4. How about cross sensitivity to temperature?

5. Have the authors tested different batches of the liquid? How can they calibrate their sensor?

6. Explaining the numerical aperture concept is not necessary. I would avoid evident explanations in the domain

7. The quality of the figures could be improved, specially Fig. 3 

8. In the introduction they mention different techniques for sensing, such as LbL. They should include references

Author Response

(The authors gave the same response as above.)

Round 2

Reviewer 2 Report

Comments and Suggestions for Authors

Authors have answered to my questions satisfactorily. In the conclusions I suggest authors could mention there is no available data for the reproducibility until now, but they will work in the future towards it.

Author Response

Hello, 

Thank you once again for your valuable feedback. Your comment about the missing information regarding reproducibility in the conclusion section is greatly appreciated. It's crucial to provide readers with insights into future research directions, and addressing reproducibility will certainly contribute to this. I incorporated this aspect in the conclusion, outlining how it will enhance the robustness and reliability of our findings.

Your input has been instrumental in refining the manuscript.

Thank you again.

Kind regards

The author
